# Real-Time Analysis of Individual Ebola Virus Glycoproteins Reveals Pre-Fusion, Entry-Relevant Conformational Dynamics

**DOI:** 10.3390/v12010103

**Published:** 2020-01-15

**Authors:** Natasha D. Durham, Angela R. Howard, Ramesh Govindan, Fernando Senjobe, J. Maximilian Fels, William E. Diehl, Jeremy Luban, Kartik Chandran, James B. Munro

**Affiliations:** 1Department of Microbiology and Physiological Systems, University of Massachusetts Medical School, Worcester, MA 01605, USA; ramesh.govindan@tufts.edu; 2Department of Molecular Biology and Microbiology, Tufts University School of Medicine and Sackler School of Graduate Biomedical Sciences, Boston, MA 02111, USA; angela.howard@tufts.edu (A.R.H.); senjobe@g.harvard.edu (F.S.); 3Department of Microbiology and Immunology, Albert Einstein College of Medicine, Bronx, NY 10461, USA; max.fels@gmail.com (J.M.F.); kartik.chandran@einstein.yu.edu (K.C.); 4Program in Molecular Medicine, University of Massachusetts Medical School, Worcester, MA 01605, USA; william.diehl@umassmed.edu (W.E.D.); jeremy.luban@umassmed.edu (J.L.)

**Keywords:** Ebola virus, envelope glycoprotein, conformational dynamics, single-molecule FRET, virus entry

## Abstract

The Ebola virus (EBOV) envelope glycoprotein (GP) mediates the fusion of the virion membrane with the membrane of susceptible target cells during infection. While proteolytic cleavage of GP by endosomal cathepsins and binding of the cellular receptor Niemann-Pick C1 protein (NPC1) are essential steps for virus entry, the detailed mechanisms by which these events promote membrane fusion remain unknown. Here, we applied single-molecule Förster resonance energy transfer (smFRET) imaging to investigate the structural dynamics of the EBOV GP trimeric ectodomain, and the functional transmembrane protein on the surface of pseudovirions. We show that in both contexts, pre-fusion GP is dynamic and samples multiple conformations. Removal of the glycan cap and NPC1 binding shift the conformational equilibrium, suggesting stabilization of conformations relevant to viral fusion. Furthermore, several neutralizing antibodies enrich alternative conformational states. This suggests that these antibodies neutralize EBOV by restricting access to GP conformations relevant to fusion. This work demonstrates previously unobserved dynamics of pre-fusion EBOV GP and presents a platform with heightened sensitivity to conformational changes for the study of GP function and antibody-mediated neutralization.

## 1. Introduction

Ebola virus (EBOV) disease outbreaks in West and Sub-Saharan Africa have occurred since the emergence of the virus in 1976 [1]. The 2014–2016 West African outbreak resulted in more than 28,000 cases and 11,000 fatalities, and the current outbreak in the Democratic Republic of Congo continues to claim lives. The most promising vaccine and therapeutic target for EBOV infection is the envelope glycoprotein (GP), which coordinates viral-host membrane fusion. GP exists as a trimer of GP1–GP2 heterodimers which resides on the surface of the EBOV virion [2]. GP1 mediates virus attachment and receptor binding, and GP2 promotes membrane fusion. Following attachment to the cell surface, EBOV is trafficked to late endosomes where GP1 is proteolytically cleaved by host cathepsins to remove the mucin-like domain (muc) and the glycan cap, forming GP_CL_ [3,4]. This cleavage event exposes the binding site for the host receptor for EBOV, Niemann-Pick C1 protein (NPC1). Binding of GP_CL_ to the luminal domain C of NPC1 (NPC1-C) is necessary, but not sufficient, to trigger GP-mediated fusion of the viral and endosomal membranes [5,6,7,8,9]. As the sole surface antigen of EBOV, GP is also the target of host neutralizing antibodies (nAbs), which may function to inhibit conformational changes required for membrane fusion [10].

Atomic resolution structures of the pre-fusion muc-deleted, transmembrane (TM)-deleted GP ectodomain (GPΔTM) in unliganded and antibody-bound states depict a conformation in which the fusion loop of GP2 is sequestered in a hydrophobic cleft, which spans the interface of two neighboring protomers [2,11,12,13]. Structures of uncleaved GP and GP_CL_ in complex with NPC1-C depict similar global conformations of the GP2 fusion loop with respect to GP1 [14,15]. In contrast, structures of post-fusion GP2 fragments indicate a loop-to-helix transition in the heptad repeat 1 (HR1) region of GP2, which would translate the fusion loop away from the surface of the virion and toward the target membrane, similar to that described for influenza hemagglutinin [16]. In the post-fusion conformation of GP, HR1 ultimately forms a single helix antiparallel to HR2 [17,18]. Thus, structural models indicate the endpoint conformations that GP adopts during membrane fusion, implying that GP is capable of undergoing large-scale conformational changes. However, direct evidence for the trajectory connecting the pre- and post-fusion conformations, including the significance of GP cleavage and NPC-1 binding in promoting this conformational rearrangement, is currently lacking.

Here, we sought to visualize the conformational dynamics of GP during the steps preceding membrane fusion. We first designed a single-molecule Förster resonance energy transfer (smFRET) imaging approach to detect real-time changes in the conformation of the trimeric GP ectodomain (GPΔTM), as well as functional, pseudovirus-associated GP (GPΔmuc) with an intact transmembrane domain. We found that both GPΔTM and GPΔmuc are intrinsically dynamic. GPΔTM exhibits a predominant conformational state that is consistent with existing structures determined by x-ray [2,11]. This conformation is also sampled by pseudovirion-associated GPΔmuc. Removal of the glycan cap of GPΔmuc and NPC1 binding destabilize the conformation depicted by crystallography, suggestive of promotion of conformations on pathway to fusion. Thus, glycan cap removal may play an active role in promoting EBOV entry beyond exposing the receptor-binding site. Finally, we also find that neutralizing antibodies bound to GPΔmuc enrich for alternative conformations, thus destabilizing access to this entry-relevant conformation, which may thereby block virus entry.

## 2. Materials and Methods

### 2.1. Cell Lines

HEK293T cells (American Type Culture Collection, Manassas, VA, USA; ATCC) were maintained at 37 °C and 5% CO_2_ in DMEM complete, consisting of Dulbecco’s Modified Eagle medium (DMEM; ThermoFisher, Waltham, MA, USA) with 10% BenchMark Fetal Bovine Serum (Gemini Bio-Products, West Sacramento, CA, USA), 1% L-Glutamine (Thermo Fisher, Waltham, MA, USA), and 1% penicillin-streptomycin (Thermo Fisher, Waltham, MA, USA). FreeStyle 293-F cells (Thermo Fisher, Waltham, MA, USA) were cultured in FreeStyle 293 Expression Medium (Thermo Fisher, Waltham, MA, USA) at 37 °C and 8% CO_2_.

### 2.2. Plasmids

The EBOV GPΔmuc Mayinga variant (Genbank accession number NP_066246) was used in all pseudoparticle experiments. GPΔmuc lacks residues 309–489 corresponding to the mucin-like domain. EBOV GPΔTM is a derivative of the GPΔmuc Mayinga variant with deletions of residues 313–463 of the mucin domain and residues 633–676 of the transmembrane domain, and encodes a C-terminal foldon trimerization peptide and 6X His tag. For single-molecule FRET experiments, GPΔTM and GPΔmuc were modified such that the nucleotide sequence encoding A1 and A4 peptides were inserted at the N-terminal positions of GP1 and GP2, respectively. A1 was inserted between amino acid 32 and 33 of GP1 (Figure 1b). A4 was inserted between amino acid 501 and 502 of GP2 (Figure 1b). pNL4.3R-E- (NIH AIDS Reagent Program, Division of AIDS, NIAID, NIH from Dr. Nathaniel Landau) was used to generate retroviral pseudoparticles, as described below. Plasmids encoding luminal domain C of NPC1 [6], and KZ52 [19] have been previously described.

### 2.3. Protein Production and Purification

Soluble NPC1-C containing an N-terminal FLAG and 6X His tag were produced by transfection in FreeStyle 293-F cells (Thermo Fisher, Waltham, MA, USA) with polyethyleneimine (PEI MAX, Polysciences, Warrington, PA, USA). Six days post-transfection, cell culture supernatant containing soluble protein was harvested and purified using PerfectPro Ni-NTA Agarose beads (PRIME GmbH, Neuss, Germany). Purified NPC1-C was dialyzed 20 mM Tris-HCl, 100 mM NaCl, 2 mM 2-mercaptoethanol, and 10% glycerol and concentrated in Vivaspin 6 ultracentrifugation spin columns (Sartorius, Gottingen, Germany). NPC1-C was stored at −80 °C prior to use. KZ52 was produced by transfection of plasmids encoding heavy and light chains into FreeStyle 293-F cells with PEI. At 3 days post-transfection, the cell culture supernatant was collected and replaced with fresh medium. The supernatant was collected again at 6 days post-transfection. The 3- and 6-day supernatant fractions were pooled, and KZ52 antibody was extracted using standard protein A affinity chromatography. After eluting from protein A in 0.2 M citric acid pH3, KZ52 was dialyzed into PBS and stored at −80 °C prior to use. GPΔTM was produced as described below.

### 2.4. Infectivity and Neutralization Assays

Retroviral pseudoparticles were generated with all tagged GPΔmuc 32-A1/501-A4 or all wild-type GPΔmuc by transfection of HEK293T cells with Lipofectamine 3000 Reagent (Thermo Fisher, Waltham, MA, USA) at 1:5 GP-encoding plasmid:pNL4-3.R-E- backbone. Virus was harvested 48 h after transfection, concentrated 10- or 20-fold with Lenti-X Concentrator (Clontech, Mountain View, CA, USA) and resuspended in DMEM complete. The infectivity or neutralization of retroviral pseudovirions was determined by Firefly Luciferase reporter gene activity using the Luciferase Assay System (Promega, Madison, WI, USA). HEK293T cells were seeded 24 h before infection in an opaque 96-well clear bottom white microplate (Thermo Fisher, Waltham, MA, USA). For infectivity assays, cells were infected directly with pseudovirions. For neutralization assays, virus was incubated with 15 µg/mL of antibody (unless otherwise stated) at 37 °C for 30 min before addition of the virus-antibody mixture to cells. Cell culture media was replaced 24 h after infection. 48 h post-infection, cells were washed with PBS (Thermo Fisher, Waltham, MA, USA), lysed with Cell Culture Lysis Reagent (Promega, Madison, WI, USA) and firefly luciferase reporter gene activity was detected using a Synergy H1 microplate reader (BioTek Instruments, Winooski, VT, USA). Infectivity of rVSV-GPΔmuc was assayed as previously described [3].

### 2.5. sNPC1-C ELISA

Pseudovirions were produced as described above by co-transfection of plasmids encoding HIV-1 backbone and either GPΔmuc or GPΔmuc 32-A1/501-A4, at a 1:1 ratio. Immediately before use, pseudoparticles were concentrated and treated for 1 h at 37 °C with 0.25 mg/mL thermolysin (Promega, Madison, WI, USA) in 50 mM Tris-HCl pH8, 0.5 mM CaCl_2_. The uncleaved control was incubated in the same way in the absence of thermolysin. Reactions were terminated with EDTA. ELISAs to determine NPC1-C binding to GPΔmuc or GP_CL_ were preformed based on previously published methods [6,20]. Briefly, high-binding 96-well ELISA plates (Thermo Fisher, Waltham, MA, USA) were coated with concentrated pseudoparticles diluted in PBS. Plates were blocked with 2% milk containing 3% bovine serum albumin in PBS. Unbound virus was washed off and 10-fold dilutions of NPC1-C (0–40 µg/mL) were added. Bound NPC1-C was detected with the horseradish peroxidase-conjugated anti-FLAG antibody (Millipore Sigma, Burlington, MA, USA) followed by Ultra-TMB substrate (Thermo Fisher, Waltham, MA, USA). All incubations were performed for 1 h at 37 °C.

### 2.6. Immunoblots

Denatured samples containing pseudovirions were run on 10% polyacrylamide gels (Millipore Sigma, Burlington, MA, USA) and transferred to nitrocellulose membranes. Membranes were blocked overnight at 4 °C. The mAb H3C8 [21] was used to detect GP and GP_CL_, and the mAb B1217M (GeneTex, Irvine, Ca, USA) to detect the HIV-1 structural protein p24. All primary antibodies were detected with a horseradish peroxidase-conjugated rabbit anti-mouse IgG Fc (Thermo Fisher, Waltham, MA, USA).

### 2.7. GPΔTM Production and Labeling for smFRET Imaging

GPΔTM was produced by transfection in FreeStyle 293-F cells (Thermo Fisher, Waltham, MA, USA) with polyethyleneimine (PEI MAX, Polysciences, Warrington, PA, USA). Wild-type GPΔTM and GPΔTM 32-A1/501-A4 expression constructs were co-transfected at a ratio of 2:1. At 6 days post-transfection, cell culture supernatant containing soluble protein was harvested and purified using PerfectPro Ni-NTA Agarose beads (PRIME GmbH, Neuss, Germany). Purified GPΔTM was exchanged into PBS supplemented with 10mM MgOAc using Vivaspin 6 ultracentrifugation spin columns (Sartorius, Gottingen, Germany) and then incubated overnight at room temperature with 5 µM each of fluorophores LD550 and LD650 (Lumidyne Technologies, New York, NY, USA) conjugated to coenzyme A (CoA; Millipore Sigma, Burlington, MA, USA) and 5µM of the labeling enzyme Acyl carrier protein synthase (AcpS). Labeled protein was purified away from free dye and enzyme via size exclusion chromatography, on a Superdex 200 Increase 10/300 GL column (GE Healthcare, Chicago, IL, USA) in PBS. Fractions containing GPΔTM were pooled and concentrated to 1–2 mg/mL. Aliquots were flash-frozen in liquid nitrogen and stored at −80 °C until use.

### 2.8. Pseudovirus Production and Labeling for smFRET Imaging

Retroviral pseudoparticles containing a single fluorescently labeled GP protomer per virion were generated by transfection of HEK293T cells with Lipofectamine 3000 Reagent (Thermo Fisher, Waltham, MA, USA) at a 5:1 ratio of pNL.4-3R-E- over plasmid encoding GP. Plasmid encoding GP consisted of a 50:1 mixture of plasmid encoding wild-type GPΔmuc:GPΔmuc 32-A1/501-A4. Virus was harvested 48 h after transfection, concentrated and labeled with fluorophores as described [22]. Briefly, harvested virus was concentrated by ultracentrifugation through a 10% sucrose cushion for 2 h at 25,000× *g*, resuspended in labeling buffer (50 mM HEPES pH 7.5, 10 mM MgOAc) and incubated overnight with 0.5 µM each of fluorophores LD550-CoA and LD650-CoA and 5 µM of the labeling enzyme AcpS. To facilitate surface-immobilization of labeled virions for imaging, DSPE-PEG2000-biotin (Avanti Polar Lipids, Alabaster, AL, USA) lipid was added to samples and incubated at room temperature for 30 min. Labeled virus was purified from unbound fluorophores and lipid by ultracentrifugation through a 6–30% OptiPrep (Millipore Sigma, Burlington, MA, USA) density gradient for 1 h at 35,000× *g*, and virus-containing fractions were identified by immunoblot for GP and p24.

### 2.9. smFRET Imaging

Purified and labeled GPΔTM or pseudovirus was immobilized and imaged using wide-field prism-based TIRF, as described [23]. smFRET data were acquired at room temperature using Metamorph software (Molecular Devices, San Jose, CA, USA) for 40 s at 25 frames/s. To generate pseudovirions containing GP_CL_, virus was treated with thermolysin as described above, terminated with phosphoramidon, and either imaged immediately or incubated with 1.7 µM (80 µg/mL) purified NPC1-C for 30 min at room temperature immediately prior to surface immobilization and imaging. For conditions containing monoclonal antibody, the indicated antibody was diluted in imaging buffer and incubated with surface immobilized pseudovirions for 30 min at room temperature immediately prior to imaging. KZ52 was prepared as described above. Purified c2G4 and c4G7 were provided by Mapp Biopharmaceutical (San Diego, CA, USA). Purified ADI-15946, ADI-15878, ADI-15742, ADI-15750, and ADI-16061 were provided by Adimab (Lebanon, NH, USA).

### 2.10. Analysis of smFRET Data

All smFRET data was processed and analyzed using the SPARTAN software package (https://www.scottcblanchardlab.com/software) in Matlab (Mathworks, Natick, MA, USA) [24]. smFRET trajectories were identified according to four criteria: (i) exceeded a minimum total fluorescence intensity; (ii) duration of smFRET trajectory exceeded 15 frames; (iii) correlation coefficient calculated from the donor and acceptor fluorescence trajectories was less than −0.1; and (iv) signal-to-noise ratio was greater than 8. Due to greater heterogeneity in the smFRET data acquired on pseudovirion-associated GP, trajectories that passed these criteria where then verified manually. Hidden Markov model (HMM) selection was performed using the Baum-Welch algorithm, with correction of the maximized log-likelihood for different numbers of model parameters using the Akaike information criterium (AIC) [25]. smFRET trajectories were idealized to the determined model using the segmental k-means algorithm [26] implemented in SPARTAN.

### 2.11. Molecular Dynamics Simulation of Fluorescently Labeled GPΔTM

Atomic coordinates were obtained from a structure of GPΔTM determined by X-ray crystallography (PDB accession 5JQ3) [11]. Amino acids corresponding to the A1 (GDSLDMLEWSLM) and A4 (DSLDMLEW) peptides were attached to the N termini of GP1 and GP2, respectively in PyMol (version 2.3.1, Schrödinger, Cambridge, MA, USA). Atomic models of LD550- and LD650-maleimide with a phosphopantetheinyl linker were constructed in PyMol (verson 2.3.1, Schrödinger, Cambridge, MA, USA). The fluorophore geometries were initially optimized at the AM1 level of theory with the sqm program in AmberTools (version 16, San Francisco, CA, USA). The geometries were further optimized, and the electrostatic potential (ESP) calculations were performed at the HF/6-31G (d) level of theory in Gaussian 9 (Gaussian, Inc., Wallingford, CT, USA). Partial atomic charges were then derived by restrained electrostatic potential (RESP) fitting using antechamber in AmberTools. Atom types and bonded parameters from the Generalized Amber Force Field (GAFF2) were assigned using antechamber and parmchk2 in AmberTools. The fluorophores and linkers were attached at the serine residue at positions 3 and 2 of the A1 and A4 peptides, respectively in LEaP. The protein component of the system was parameterized with the Amber force field (ff14SB). The entire system was charge-neutralized and solvated in explicit water using the TIP3P model with periodic boundary conditions in LEaP. The system was energy minimized for 0.1 ns, followed by a 50 ns simulation run in the NPT ensemble, with temperature and pressure maintained at 300 K and 1 atm through the use of Langevin dynamics and the Nose’-Hoover Langevin piston method, respectively. Simulations were run using NAMD (version 2.12, Urbana-Champaign, IL, USA) on the Tufts Research Cluster.

## 3. Results

### 3.1. Site-Specific Fluorescent Labelling of EBOV GP

The application of smFRET imaging requires site-specific attachment of donor and acceptor fluorophores to GP. We specifically sought to probe movement of GP1 with respect to GP2, as well as movement of the viral fusion loop that is proximal to the N terminus of GP2. Both motions are predicted to be critical to the mechanism of GP-mediated membrane fusion [16]. To this end, we modified GPΔTM from the Mayinga strain of EBOV for the site-specific attachment of donor and acceptor fluorophores (Figure 1a). This construct contained deletions of the mucin domain and transmembrane domain, and included a foldon trimerization peptide and C-terminal 6X His tag, similar to that characterized by x-ray crystallography but with residue T at position 42 to maintain the native glycosylation site [11]. We inserted the 12-residue A1 peptide into the N terminus of GP1, and the 8-residue A4 peptide into the N-terminus of GP2 to generate GPΔTM 32-A1/501-A4 (Figure 1b,c). These peptide tags enable enzymatic attachment of fluorophore conjugates to a specific serine residue within each peptide [27,28]. Molecular dynamics simulation of GPΔTM in the pre-fusion conformation with the peptide tags and conjugated fluorophores suggested an inter-fluorophore distance of approximately 30 Å, predictive of high FRET (Figure 1c).

We next verified that these peptide-insertion sites did not disrupt GP function by creating analogous insertions in full-length, transmembrane-intact GPΔmuc (i.e., GPΔmuc 32-A1/501-A4). When expressed either GPΔmuc 32-A1/501-A4 or untagged GPΔmuc on pseudovirions using a recombinant vesicular stomatitis virus (rVSV) core that encodes GFP [24]. Pseudovirions containing GPΔmuc 32-A1/501-A4 maintained infectivity levels comparable to those bearing untagged GPΔmuc (Figure 2a). Since the low density of glycoproteins on the surface of retroviral particles provides the best platform for smFRET imaging, we formed HIV-1 particles pseudotyped with GPΔmuc 32-A1/501-A4 and characterized incorporation into virus particles, cleavage of the glycan cap by thermolysin, sensitivity to neutralization by a panel of antibodies, and NPC1-C binding (Figure 2b–d). In all cases, pseudovirion-associated GPΔmuc 32-A1/501-A4 maintained phenotypes comparable to wild-type GPΔmuc.

### 3.2. Conformational Dynamics of EBOV GPΔTM

To generate trimeric protein for smFRET imaging, 293-Freestyle cells were co-transfected with plasmids encoding GPΔTM 32-A1/501-A4 and untagged GPΔTM at a ratio of 1:2 (Section 2). This was found to optimize the proportion of GPΔTM trimers containing a single GPΔTM 32-A1/501-A4 protomer. smFRET measurements could then be made that originated from a single donor-acceptor fluorophore pair on the same GPΔTM protomer (Figure 3a), as previously described for studies of other envelope glycoproteins on the surface of virions [22,23]. Trimers containing enzymatically attached fluorophores were surface immobilized and imaged by prism-based total internal reflection fluorescence (TIRF) microscopy (Section 2) (Figure 3a). We confirmed by western blot that the sample of GPΔTM used in smFRET imaging contained minimal amounts of GP0 and remained intact during protein expression and purification (Figure 3b). We observed smFRET trajectories indicating intrinsic and spontaneous fluctuations between multiple FRET states during the observation period of approximately 20 s (Figure 3c). Hidden Markov modeling (HMM) indicated a predominant high-FRET state (0.74 ± 0.08) with 52.6% occupancy, as displayed in histograms of the FRET values compiled from the population of observed molecules (Figure 3d, left). This high-FRET state is consistent with the ~30 Å inter-fluorophore distance predicted by molecular dynamics simulation of fluorescently labeled GP in the conformation determined by x-ray crystallography (Figure 1c) [2,11,14]. We also observed two additional states, 0.27 ± 0.08 and 0.50 ± 0.08 FRET, suggestive of alternative GP conformations in which the distance between the fluorophore pair increases. The intermediate- and low-FRET states were present with occupancies of 26.9% and 20.5%, respectively (Table 1). A transition density plot (TDP) displays the relative frequencies of observed transitions within this population of trimers. The TDP shows roughly equal transition frequencies between the high- and intermediate-FRET states, and between the intermediate- and low-FRET states; transitions directly between the high- and low-FRET states were rarely seen (Figure 3d, right). The TDP is symmetric about the diagonal axis, which indicates equilibrium dynamics between multiple states. These data suggest that pre-fusion GPΔTM is intrinsically dynamic, sampling two conformations in addition to that which has been described by crystallography.

### 3.3. Conformational Dynamics of Pseudovirion-Associated EBOV GPΔmuc

We next asked whether the dynamics observed for GPΔTM are representative of full-length functional GPΔmuc. We therefore transferred the same fluorophore attachment strategy to GPΔmuc on the surface of retroviral pseudovirions. As done previously for influenza hemagglutinin and HIV Env [22,23], we formed pseudovirions with the HIV core and an excess of wild-type GPΔmuc over GPΔmuc 32-A1/501-A4 (Section 2). This dilution of GPΔmuc 32-A1/501-A4 with wild-type GPΔmuc maximized the probability that only a single fluorophore pair was present on the surface of each virion (Figure 4a). Similar to the data for GPΔTM, we observed smFRET trajectories indicating intrinsic and spontaneous fluctuations between multiple FRET states (Figure 4b). Notably, HMM analysis of the smFRET trajectories revealed the same low- (0.28 ± 0.07), intermediate- (0.50 ± 0.08) and high- (0.73 ± 0.08) FRET states as observed for GPΔTM (Figure 3c,d). However, we observed lower intrinsic occupancy of the high-FRET state, higher occupancy in the low-FRET state, and slightly lower occupancy of the intermediate-FRET state as compared to GPΔTM (Table 1). The TDP indicates that transitions occur between all three FRET states, with transitions between high and intermediate FRET, and between intermediate and low FRET being most frequent (Figure 4b). These data suggest that pre-fusion, unliganded GPΔmuc, similarly to GPΔTM, can spontaneously and reversibly transition between multiple conformations on the surface of pseudovirions.

### 3.4. Effects of Proteolytic Cleavage and NPC1 Binding on EBOV GPΔmuc Conformation

We next examined whether proteolytic removal of the glycan cap and subsequent binding to soluble NPC1-C (sNPC1-C) induce conformational changes in GPΔmuc. We treated the pseudovirions with thermolysin to generate GP_CL_ immediately prior to smFRET imaging (Section 2). Individual traces, as well as the population FRET histogram, showed increased occupancy in the intermediate-FRET state for GP_CL_ as compared to GPΔmuc (Figure 4c). Also notable from the TDP was a decrease in the overall level of dynamics, indicated by reduced probability density for all observed transitions. This reflects a reduction in the number of transitions between each FRET state compared to GPΔmuc (Figure 4b). Again, the observed transitions occurred primarily between the intermediate- and high-FRET states and were rarely detected between the low- and high-FRET states. We then asked whether NPC1 binding might further remodel GP conformation. We observed that addition of sNPC1-C to GP_CL_ further promoted an intermediate-FRET state and reduced the occupancy of the low- and high-FRET states (Figure 4d). Equally as dramatic was the reduction in the level of overall dynamics, as depicted in the TDP. These data demonstrated that proteolytic removal of the glycan cap destabilizes the high-FRET conformation preferred by uncleaved GPΔmuc. NPC1 binding further destabilizes this state.

### 3.5. Neutralizing Antibodies Mediate GP Conformation

We next asked whether nAbs might function by remodeling GP in a manner that is distinct from proteolytic cleavage and NPC1 binding. In particular, antibodies that interact with GP proximal to the fusion loop are predicted to function by preventing conformational changes necessary for membrane fusion. The antibody KZ52 showed a modest concentration-dependent stabilization of the low-FRET state, which was most pronounced at the saturating concentration of 500 µg/mL (Figure 5a). This mild effect on GP conformation is consistent with reports of KZ52 incompletely neutralizing EBOV [19,29]; KZ52 may also function by limiting proteolytic cleavage by cathepsins [30]. Additional nAbs were tested at 15 µg/mL, a concentration that yields minimally 90% neutralization (Figure 2b). The base-binding antibodies c2G4 and c4G7 [31,32] also stabilized the low-FRET state, to a greater extent than KZ52 (Figure 5a). In contrast, ADI-15946, an antibody that competes with KZ52 but bridges the GP1 core and glycan cap [33,34,35], modestly stabilized the high-FRET state. We also tested nAbs ADI-15878 and ADI-15742, which contact the GP2 fusion loop and adjacent GP1 protomer [33,34,36]. These nAbs significantly increased the occupancy in the high-FRET state (Figure 5b). The glycan-cap binding nAb ADI-15750 slightly stabilized the low-FRET state (Figure 5c) [33]. Finally, the GP2 stalk-binding nAb ADI-16061 (Figure 5d) showed no detectable change in FRET-state occupancy compared to unliganded GP, indicating that the current fluorophore placement does not report on its mechanism of action [33]. In all cases, transitions to all three FRET states persisted as indicated by the TDPs. Notably, all the nAbs tested stabilized states distinct from proteolytic cleavage and NPC1-C binding. This suggests that their mechanisms of action may be to inhibit transition to the intermediate-FRET conformation.

## 4. Discussion

The spring-loaded mechanism of type-I viral fusogens, based largely on studies of influenza hemagglutinin, predicts that dissociation of GP1 from GP2 permits refolding of GP2, which moves the fusion loop toward the target membrane [37]. Structures of pre- and post-fusion GP support this model [2,11,14,18]. The site-specific placement of fluorophores at the N termini of GP1 and GP2 allowed us to visualize movement of the N terminus of GP2, which is proximal to the fusion loop, and the movement of GP1 with respect to GP2. We therefore anticipate that the dynamics observed here reflect conformational changes in GP that relate to its mechanism of membrane fusion. Our approach compliments previous studies of GP by accounting for several key features: (i) real-time changes in GP conformation on the timescale of 10–100 milliseconds; (ii) the use of GPΔTM as well as GPΔmuc with an intact transmembrane and cytoplasmic region, which permits their direct comparison; (iii) the functionally relevant context of a viral membrane; and (iv) measurements on single trimers, which precludes the need for averaging measurements from a population of asynchronous trimers. Our results indicate that GP samples multiple conformations beyond what has been described in structural investigations. Molecular modeling supports the identification of the high-FRET state as the conformation depicted in the crystal structures of GP, with the fusion loop positioned in a hydrophobic cleft and contacting the neighboring GP promoter. While the high-FRET state has the highest occupancy for both GPΔTM and GPΔmuc, our comparison indicates that the presence of the transmembrane domain shifts the conformational equilibrium away from the conformation described by crystallography and favors transition to an alternative conformation.

In order to fully capture the conformational dynamics of complex macromolecules, multiple smFRET signals, stemming from distinct fluorophore attachment strategies are necessary. A parallel study used smFRET imaging with both fluorophores attached to GP2 and was thus insensitive to inter-domain movements [38]. This study indicated no conformational change upon cleavage of the glycan cap. This suggests that the conformational shift toward intermediate FRET seen upon glycan cap cleavage in the present study mainly reflects repositioning of GP1 with respect to GP2. This rearrangement may aid in priming GP1 for interaction with NPC1, and in facilitating other downstream events related to fusion. Recent structural modeling suggests that movement of the fusion loop away from the surface of the virion would require outward motion of GP1, but not necessarily GP1 release, in order to allow the fusion loop access to the space above the GP core [39]. Our results confirm that GP1 does not dissociate from GP2 at this stage, since we still observe reversible transitions into and out of intermediate FRET. Removal of the glycan cap may therefore relieve a physical restriction on the movement of GP2 [40], allowing the fusion loop greater mobility. Das et al. also reported a dramatic repositioning of the GP2 N terminus upon sNPC1-C binding [38]. Thus, the effect of glycan cap cleavage seen in the present study may potentiate GP2 rearrangements promoted by NPC1 binding. The further stabilization of intermediate FRET seen here upon sNPC1-C binding would then reflect additional conformational rearrangements in GP1 and GP2, which result in no detectable change in FRET with the current fluorophore attachment strategy and signal-to-noise ratio. However, when the results of the present study and those of Das et al. are interpreted together, a more complete understanding emerges of the independent effects of glycan cap cleavage and NPC1 binding, respectively. In summary, the current data support the idea that the glycan cap plays an essential role beyond obscuring the receptor-binding site. Specifically, the glycan cap mediates the pre-fusion conformational dynamics of GP, limiting transition to conformations that are putatively on pathway to fusion. An intriguing possibility is that the glycan cap serves to prevent spontaneous triggering of GP prior to arrival of EBOV in the late endosome, at which point the glycan cap is removed by the cellular cathepsins.

In contrast to glycan cap removal and sNPC1-C binding, the nAbs tested here that alter GP conformation either enrich the low-FRET (KZ52, c2G4, and c4G7) or high-FRET (ADI-15878 and ADI-15742) states. We cannot exclude the possibility that base-binding nAbs may cause local changes in the regions of fluorophore attachment. Therefore, it remains possible that the high- and low-FRET states result from local changes in conformation or fluorophore orientation rather than global changes in GP conformation. In either case, these data support the idea that the mechanism of neutralization by these nAbs is to inhibit transition to conformations, such as the intermediate-FRET state, that are relevant to the fusion reaction [10]. At present we cannot rule out the possibility that the neutralizing antibodies enrich for conformations that are distinct from those seen for unliganded and receptor-bound GP, but which are not distinguishable using the current fluorophore placement. The use of additional fluorophore attachment sites in GP1 and GP2, as well as unbiased molecular modeling and structural techniques will further resolve conformations of GP in bound and unliganded states without relying on pre-existing conformations of GP that are favored under conditions of crystal formation. smFRET imaging provides a unique platform for evaluating GP ligands and mutant GP variants that stably present antigenic conformations and should thus find utility in future studies designing novel vaccine candidates and antiviral therapeutics that target GP. Taken together, these observations demonstrate that the activity of GP in membrane fusion is regulated by ligands that modulate the intrinsic conformational dynamics of GP. smFRET imaging provides a unique platform for evaluating GP ligands and mutant GP variants that stably present antigenic conformations and should thus find utility in future studies designing novel vaccine candidates and antiviral therapeutics that target GP.

## Figures and Tables

**Figure 1 viruses-12-00103-f001:**
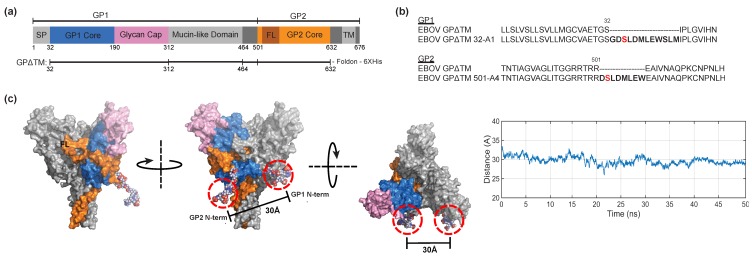
Site-specific fluorescent labelling of EBOV GP for smFRET. (**a**) Structural organization of GPΔTM. Adapted from Ref. [11]. FL, fusion loop; SP, signal peptide; TM, transmembrane helix. (**b**) EBOV GPΔTM and GPΔmuc were modified to contain peptide tags (bold) in GP1, between amino acid position 32 and 33, and in GP2, between amino acid positions 501 and 502, to facilitate enzymatic labelling with fluorophores at the serine residues highlighted in red. (**c**) Surface rendering of EBOV GPΔTM (derived from PDB accession 5JQ3 [11]) with fluorophores attached. MD simulation of fluorescently labeled GPΔTM 32-A1/501-A4 predicts an inter-fluorophore distance of approximately 30 Å. After minimization and equilibration, the time-averaged distance between the centers of mass of the fluorophores was determined through a 50-ns simulation (see Section 2).

**Figure 2 viruses-12-00103-f002:**
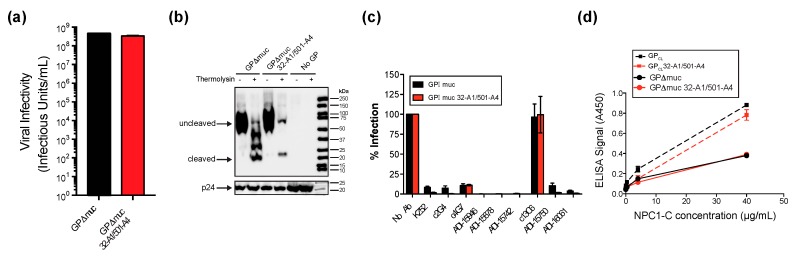
Characterization of pseudovirions containing peptide tags for site-specific fluorophore attachment. (**a**) Infectivity of pseudovirions containing either GPΔmuc or GPΔmuc 32-A1/501-A4 using a recombinant vesicular stomatitis virus (VSV) core that encodes GFP. Viral infectivity was determined by counting eGFP-positive Vero cells 14–16 h post infection. Results are shown as the mean ± standard error (n = 3). (**b**) Western blot of retroviral pseudovirions containing GPΔmuc, GPΔmuc 32-A1/501-A4 or no GP after 1 h mock treatment (−) or treatment with thermolysin (+) to cleave the glycan cap and generate GP_CL_. The HIV structural protein p24 indicates the total amount of virus loaded. (**c**) Neutralization of retroviral pseudoparticles by monoclonal antibodies. Antibodies were tested at 15 µg/mL for neutralizing activity against retroviral pseudoparticles containing either GPΔmuc (black bars) or GPΔmuc 32-A1/501-A4 (red bars). Luciferase activity was detected 48 h post-infection and expressed as a percentage of infection with no antibody (No Ab). Results are shown as the mean ± standard error (n = 2). (**d**) ELISA for NPC1-C binding to retroviral pseudovirions containing only GPΔmuc or GPΔmuc 32-A1/501-A4, with or without thermolysin treatment to generate GP_CL_ or GP_CL_ 32-A1/501-A4. Results are shown as the mean ± standard error (n = 3).

**Figure 3 viruses-12-00103-f003:**
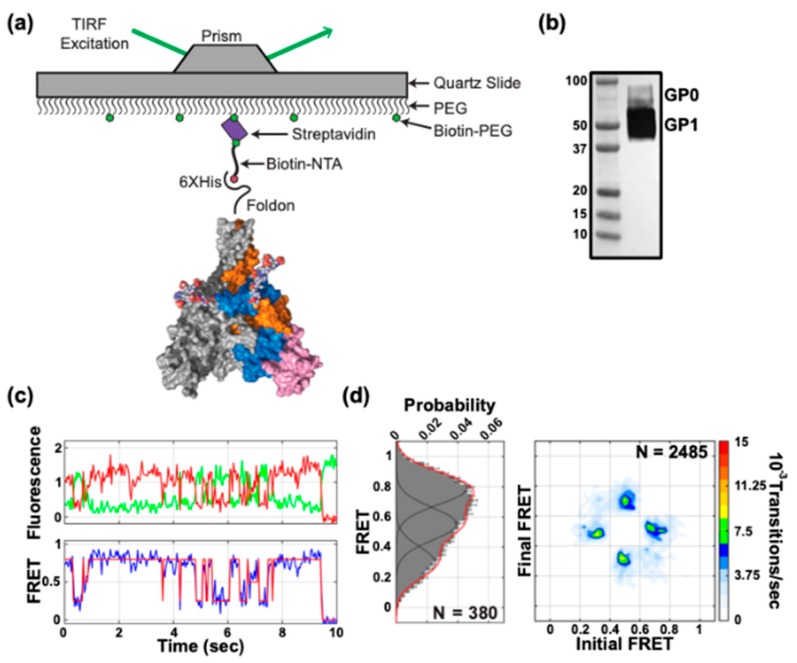
Conformational dynamics of EBOV GPΔTM. (**a**) smFRET imaging of GPΔTM trimers containing a single fluorescently-labelled protomer within an otherwise wild-type GPΔTM trimer were surface-immobilized via biotin-NTA on a streptavidin-coated quartz microscope slide for smFRET imaging with TIRF microscopy. (**b**) Western blot of purified GPΔTM used in smFRET imaging using an anti-GP1 antibody (H3C8). (**c**) Representative fluorescence trace (top) showing fluorescence intensity over time for a donor (LD550; green) and acceptor (LD650; red) fluorophore pair on a single GPΔTM trimer, and the corresponding FRET trajectory (bottom). Overlaid on the FRET trajectory is the idealization (red) resulting from HMM analysis. (**d**) (Left) Population FRET histogram for GPΔTM trimers composed of FRET trajectories summed over the observation window. Overlaid on the histograms are Gaussian distributions with means 0.3, 0.5 and 0.7 that reflect the results of HMM analysis. N indicates the number of trimers in the population histogram. (Right) Transition density plot (TDP) displaying the relative frequencies of observed transitions generated from idealizations of individual FRET trajectories. N indicates the total number of transitions within the population of trimers analyzed.

**Figure 4 viruses-12-00103-f004:**
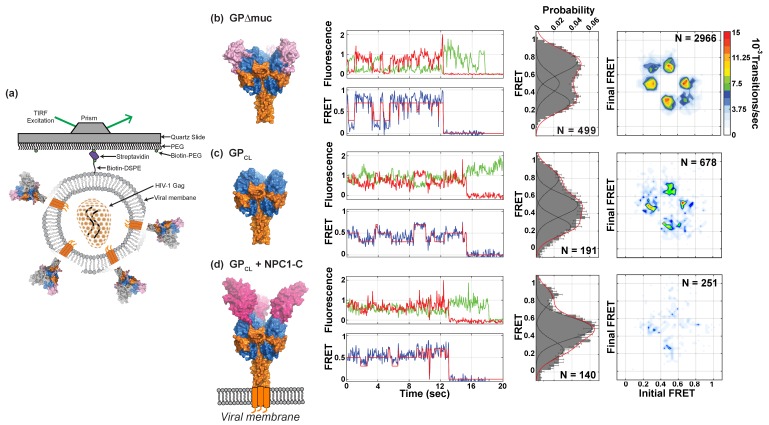
Proteolytic priming and NPC1 binding to EBOV GPΔmuc pseudovirions promote a pre-existing conformation. (**a**) Pseudovirions containing a single fluorescently labelled protomer within an otherwise native GPΔmuc trimer were surface-immobilized on streptavidin-coated quartz slides for smFRET imaging with TIRF microscopy. (**b**) (Left) Structure of EBOV GPΔmuc, with subdomains labelled as in Figure 1c. (Left/center) Representative fluorescence trace (top) showing fluorescence intensities over time for a donor (LD550; green) and acceptor (LD650; red) fluorophore pair on a single GPΔmuc on the surface of a pseudovirion. The corresponding FRET trajectory (bottom) is shown in blue. Overlaid on the FRET trajectory is the idealization (red) resulting from HMM analysis. (Right/center) Population FRET histogram for GPΔmuc trimers on pseudovirions composed of FRET trajectories summed over time. Overlaid on the histograms are Gaussian distributions with means 0.3, 0.5, and 0.7 that reflect the results of HMM analysis. N indicates the number of trimers in the population histogram. (Right) TDPs display the relative frequencies of observed transitions, generated from idealizations of individual FRET trajectories for the GPΔmuc population. N indicates the total number of transitions within the population of trimers analyzed. (**c**) Structural model and smFRET data, displayed as in (**b**), for GPΔmuc treated with thermolysin to remove the glycan cap (GP_CL_). (**d**) Structural models and smFRET data for GP_CL_ bound to sNPC1-C. Structural models were adapted from PDB accessions 5JQ3 and 5F1B [11,14]. Approximate location of the viral membrane is shown.

**Figure 5 viruses-12-00103-f005:**
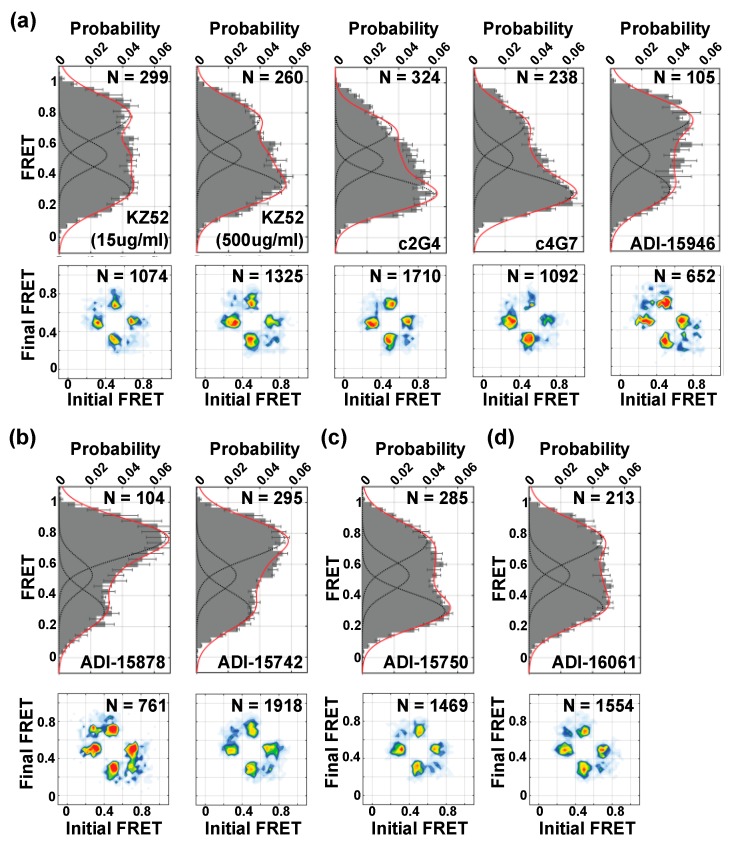
Neutralizing antibodies stabilize distinct EBOV GPΔmuc conformations. Population FRET histograms and corresponding TDPs, displayed as in Figure 3 and Figure 4, for EBOV GPΔmuc in the presence of 15 µg/mL (unless otherwise noted) of the following: (**a**) Base-binding antibodies KZ52, c2G4, c4G7, and ADI-15946; (**b**) base-binding antibodies that contact the GP2 fusion loop and adjacent GP1 protomer ADI-15878 and ADI-15742; (**c**) Glycan-cap-binding antibody ADI-15750; and (**d**) GP2 stalk-binding antibody ADI-16061.

**Table 1 viruses-12-00103-t001:** Förster resonance energy transfer (FRET) state occupancies determined through hidden Markov modeling.

	Occupancies (%)		
	Low FRET	Intermediate FRET	High FRET
GPΔTM	20.5%	26.9%	52.6%
GPΔmuc	31.4%	23.6%	45.0%
GP_CL_	22.2%	45.8%	32.0%
GP_CL_ + NPC1-C	19.8%	53.4%	26.8%
GPΔmuc + KZ52 (15 μg/mL)	35.7%	23.3%	41.0%
GPΔmuc + KZ52 (500 μg/mL)	42.0%	23.2%	34.8%
GPΔmuc + c2G4	46.7%	24.9%	28.4%
GPΔmuc + c4G7	47.2%	25.6%	27.2%
GPΔmuc + ADI-15946	27.1%	25.0%	47.9%
GPΔmuc + ADI-15878	20.8%	20.2%	59.0%
GPΔmuc + ADI-15742	21.8%	23.3%	49.3%
GPΔmuc + ADI-15750	41.4%	23.6%	35.0%
GPΔmuc + ADI-16061	32.0%	23.1%	44.9%

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
