# Peer review of "Real-Time Analysis of Individual Ebola Virus Glycoproteins Reveals Pre-Fusion, Entry-Relevant Conformational Dynamics"

_viruses, 2020, doi:10.3390/v12010103_

Round 1
Reviewer 1 Report
In this article, Durham and colleagues employed smFRET imaging approach to investigate the structural dynamics of the EBOV GP trimeric ectodomain, and the functional transmembrane protein on the surface of pseudovirions. The authors demonstrated that pre-fusion GP is dynamic and samples multiple conformations. Proteolytic removal the glycan cap and NPC1 binding shift the conformational equilibrium. Furthermore, several neutralizing antibodies enrich alternative conformational states.
As the authors commented, studies on the dynamics of pre-fusion EBOV GP provide a platform with heightened sensitivity to conformational changes for the study of GP function and antibody-mediated neutralization.
It would be more than ready for publication if the authors would provide additional information regarding the following comments. 

 Major comment 
 About functional insight(s):
If a functional assessment, using a cell model, can be performed to evaluate and confirm the knowledge learned from smFRET, I think such additional data will increase the significance of this article.
Minor questions 
 1. As the beginning of the sentence, spell out 6 as “Six” (line 96)
 2. Please check the units used in Table 1 ( g/ml) and Fig 5A (µg/ml)
Author Response
In this article, Durham and colleagues employed smFRET imaging approach to investigate the structural dynamics of the EBOV GP trimeric ectodomain, and the functional transmembrane protein on the surface of pseudovirions. The authors demonstrated that pre-fusion GP is dynamic and samples multiple conformations. Proteolytic removal the glycan cap and NPC1 binding shift the conformational equilibrium. Furthermore, several neutralizing antibodies enrich alternative conformational states.
As the authors commented, studies on the dynamics of pre-fusion EBOV GP provide a platform with heightened sensitivity to conformational changes for the study of GP function and antibody-mediated neutralization.
It would be more than ready for publication if the authors would provide additional information regarding the following comments.
We thank the reviewer for his/her support of our work. We have tried to address each comment below.
Major comment
About functional insight(s):
If a functional assessment, using a cell model, can be performed to evaluate and confirm the knowledge learned from smFRET, I think such additional data will increase the significance of this article.
We agree with the reviewer that functional assessment of the modified GP constructs is essential. Indeed, this was the purpose of Figure 2, in which we show that tagged GP is functional in infectivity (in Vero cells), neutralization, and ELISA assays.
Several additional functional assessments, beyond the experiments provided here, are planned for future studies. These include experiments to stabilize GP conformations observed by smFRET though site-specific mutagenesis and protein engineering. We expect that these mutants would reduce or prevent GP dynamics and would therefore show little to no fusion.
Finally, our long-term goal is to observe with smFRET GP conformational changes during cell entry. Adaptation of smFRET to cellular imaging is technically challenging and will likely require significant new technology and methods development.
Minor questions
As the beginning of the sentence, spell out 6 as “Six” (line 96) 2. Please check the units used in Table 1 ( g/ml) and Fig 5A (µg/ml)
Both minor questions have been edited according to the reviewers suggestions.
Reviewer 2 Report
The report describes use of single molecule FRET (smFRET) to detect conformational states of the Ebola virus glycoprotein (GP). GP is the sole virally encoded membrane protein presented on the surface of the virion. It serves to bind cells, mediate uptake by endocytosis, bind an intra-endosomal receptor, NPC1 and then mediate membrane fusion of virus and endosomal membranes. By analogy to other virus GPs, it is expected that Ebola virus GP needs to undergo conformational changes to perform each function.
Here, smFRET is used to detect conformational states in GP in solution, on pseudotyped particles and after proteolytic activation. An interesting approach is taken to labeling the GPs with fluorescent dyes, using enzymatic tagging of specific peptides added to the GP. The impact of the adducts is well described, as is the construct of chimeric GPs and appears to not alter function for infection of cells and so likely adopts a native conformation.
It is found that the GP (after removal of the glycan cap) is able to adopt 3 states termed high, medium and low. These appear to exist in dynamic equilibrium, with each GP trimer able to move between each state at some rate. The frequency of each state changes between soluble, artificially trimerized and virion anchored, natively trimerized forms, of the GP. The post-proteolytic cleaved form of GP again has an altered distribution of states as well. While it is tempting to attribute the differences in frequency of the conformational states to function, it may be just a function of construct and one or only a couple of the states may be relevant. Using this approach, it would be interesting to examine states of known mutants of GP, particularly those that specifically block membrane fusion. Furthermore, in the discussion, it is suggested that the known crystal structures of GP might represent the high-FRET state. However, the evidence for this conclusion needs to be better supported by experimental data. It is suggested that a modeling method was applied but the details are not clearly described. This should be provided in more detail.
The use of neutralizing antibodies is interesting as each has known targets for GP and would be expected to shift the frequency of states based on binding affinity and binding site and mechanism of neutralization. Indeed, each antibody does alter the frequency of states seen. However, shifting towards low or high states appears to differ greatly between each antibody, as might be expected. It is possible that an antibody may drive alternative states that are not within the conformational space sampled by the native protein. The application of additional models might allow identification of such states. For example c4G7 appears to have an additional peak above the high state and AD-15750 appears to shift the middle peak upward. Comment should be made in the discussion as to such alternative states, if they can be distinguished by smFRET and their relevance to structural models of antibody binding in context of the limitations of analysis.
Author Response
The report describes use of single molecule FRET (smFRET) to detect conformational states of the Ebola virus glycoprotein (GP). GP is the sole virally encoded membrane protein presented on the surface of the virion. It serves to bind cells, mediate uptake by endocytosis, bind an intra-endosomal receptor, NPC1 and then mediate membrane fusion of virus and endosomal membranes. By analogy to other virus GPs, it is expected that Ebola virus GP needs to undergo conformational changes to perform each function.
Here, smFRET is used to detect conformational states in GP in solution, on pseudotyped particles and after proteolytic activation. An interesting approach is taken to labeling the GPs with fluorescent dyes, using enzymatic tagging of specific peptides added to the GP. The impact of the adducts is well described, as is the construct of chimeric GPs and appears to not alter function for infection of cells and so likely adopts a native conformation.
We thank the reviewer for his/her supportive comments on our work. We have tried to address all comments completely below.
It is found that the GP (after removal of the glycan cap) is able to adopt 3 states termed high, medium and low. These appear to exist in dynamic equilibrium, with each GP trimer able to move between each state at some rate. The frequency of each state changes between soluble, artificially trimerized and virion anchored, natively trimerized forms, of the GP. The post-proteolytic cleaved form of GP again has an altered distribution of states as well. While it is tempting to attribute the differences in frequency of the conformational states to function, it may be just a function of construct and one or only a couple of the states may be relevant. Using this approach, it would be interesting to examine states of known mutants of GP, particularly those that specifically block membrane fusion.
The reviewer is correct that examining the dynamics GP mutants with smFRET is of great interest. Indeed, we have ongoing studies of mutations that arose during the current and recent Ebola outbreaks in Africa, in addition to mutations determined in previously published works. However, since several steps seem to be required for EBOV fusion, at least one of which (the fusion trigger) is unknown, it is unclear whether known mutants are specifically blocked at the step of fusion or during one of several upstream steps. Thus, in order to interpret the smFRET data from such mutants we first need to determine at what step the fusion reaction is arrested. This too is part of our ongoing studies.
Furthermore, in the discussion, it is suggested that the known crystal structures of GP might represent the high-FRET state. However, the evidence for this conclusion needs to be better supported by experimental data. It is suggested that a modeling method was applied but the details are not clearly described. This should be provided in more detail.
The Reviewer is correct that we were remiss in omitting a section in the Materials and Methods detailing the molecular dynamics simulation used to determine the distance between the fluorophores in the GP conformation depicted in the crystal structure. A section has now been added indicating this procedure. Clarification is also added in the Results (lines 256-258) and in the Figure 1 legend. Furthermore, future experiments are planned to experimentally address this more adequately, including to engineer GP using e.g. site specific disulphide bonds to stabilize the crystal conformation.
The use of neutralizing antibodies is interesting as each has known targets for GP and would be expected to shift the frequency of states based on binding affinity and binding site and mechanism of neutralization. Indeed, each antibody does alter the frequency of states seen. However, shifting towards low or high states appears to differ greatly between each antibody, as might be expected. It is possible that an antibody may drive alternative states that are not within the conformational space sampled by the native protein. The application of additional models might allow identification of such states. For example c4G7 appears to have an additional peak above the high state and AD-15750 appears to shift the middle peak upward. Comment should be made in the discussion as to such alternative states, if they can be distinguished by smFRET and their relevance to structural models of antibody binding in context of the limitations of analysis.
The Reviewer makes an excellent point that multiple conformations could reside within a single modeled FRET state. We have acknowledged this possibility in the Discussion (lines 416-425) and eluded to future studies aimed at resolving such additional conformations.